# Beneficial Effects of Pulmonary Vasodilators on Pre-Capillary Pulmonary Hypertension in Patients with Chronic Kidney Disease on Hemodialysis

**DOI:** 10.3390/life12060780

**Published:** 2022-05-24

**Authors:** Keiji Kimuro, Kazuya Hosokawa, Kohtaro Abe, Kohei Masaki, Satomi Imakiire, Takafumi Sakamoto, Hiroyuki Tsutsui

**Affiliations:** Department of Cardiovascular Medicine, Faculty of Medical Sciences, Kyushu University, 3-1-1, Maidashi, Higashi-Ku, Fukuoka 812-8582, Japan; kkimuro@junnai.org (K.K.); abe.kotaro.232@m.kyushu-u.ac.jp (K.A.); komasaki@junnai.org (K.M.); imakiire@junnai.org (S.I.); sakamoto.takafumi.735@m.kyushu-u.ac.jp (T.S.); htsutsui@cardiol.med.kyushu-u.ac.jp (H.T.)

**Keywords:** pulmonary hypertension, pulmonary vasodilator, pre-capillary pulmonary hypertension, hemodialysis, chronic kidney disease, exercise tolerance, pulmonary hemodynamics, pulmonary edema, fluid management

## Abstract

Background: In patients with chronic kidney disease (CKD) on hemodialysis, comorbid pulmonary hypertension (PH) aggravates exercise tolerance and eventually worsens the prognosis. The treatment strategy for pre-capillary PH, including combined pre- and post-capillary PH (Cpc-PH), has not been established. Objectives: This study aimed to evaluate the impact of pulmonary vasodilators on exercise tolerance and pulmonary hemodynamics in patients with CKD on hemodialysis. Methods and Results: The medical records of 393 patients with suspected PH who underwent right heart catheterization were reviewed. Of these, seven patients had isolated pre-capillary PH and end-stage CKD on hemodialysis. Pulmonary vasodilators decreased pulmonary vascular resistance from 5.9 Wood units (interquartile range (IQR), 5.5–7.6) at baseline to 3.1 Wood units (IQR, 2.6–3.3) post-treatment (*p* = 0.02) as well as increased pulmonary capillary wedge pressure from 10 mmHg (IQR, 7–11) to 11 mmHg (IQR, 8–16) (*p* = 0.04). Pulmonary vasodilators increased the World Health Organization functional class I or II from 0% to 100% (*p* = 0.0002) and the 6 min walk distance from 273 m (IQR, 185–365) to 490 m (IQR, 470–550) (*p* = 0.03). Conclusions: Pulmonary vasodilators for PH in patients with CKD on hemodialysis decrease pulmonary vascular resistance and eventually improve exercise tolerance. Pulmonary vasodilators may help hemodialysis patients with pre-capillary PH, although careful management considering the risk of pulmonary edema is required.

## 1. Introduction

Approximately 720,000 patients in the United States and 340,000 patients in Japan are currently on maintenance hemodialysis [1,2]. Annually, 125,000 patients with chronic kidney disease (CKD) initiate hemodialysis in the United States. In Japan, hemodialysis is the standard treatment for bridging to kidney transplantation, and it is frequently used as a destination therapy for patients with end-stage kidney disease. Overall, 10–50% of patients on hemodialysis are reported to have pulmonary hypertension (PH) as a complication [3,4,5,6,7]. Comorbid PH in dialysis patients causes dysdialysis syndrome, including hemodialysis-related hypotension, resulting in inadequate fluid management and eventually leading to resistant hypertension, heart failure, arrhythmia, and sudden death.

The pathogenesis of PH associated with hemodialysis is multifactorial [8,9,10,11,12]. This type of PH may be pre-capillary (pulmonary artery hypertension), post-capillary (left heart disease), or combined pre- and post-capillary PH (Cpc-PH). Increased cardiac output caused by dialysis shunt, renal anemia, and catheter-related bacteremia induces volume overload and worsens PH [4,6]. Therefore, PH associated with hemodialysis is classified as PH with unclear multifactorial mechanisms. Post-capillary PH is the most common type of hemodialysis-associated PH, which is caused by inappropriate fluid management and left heart diseases, such as hypertensive heart disease, cardiac amyloidosis, and coronary artery disease [4,13,14]. Furthermore, pre-capillary PH occurs in 13% of the patients [13]. The dialysis shunt increases vessel shear stress; mineral and bone metabolism dysregulation causes abnormal vascular calcification. These mechanisms cause endothelial damage and microvascular remodeling in the pulmonary vasculature.

PH is an independent predictor of mortality in patients with CKD who are on hemodialysis [15,16]. The five-year survival of patients with CKD-associated PH is 40%. Moreover, the treatment strategies for PH remain unestablished; thus, PH in patients with CKD on hemodialysis is a serious problem. Pivotal treatments for post-capillary PH consist of fluid management and dedicated treatment for left ventricular dysfunction. However, there is no established treatment for pre-capillary PH. The effect of pulmonary vasodilators on symptoms, exercise tolerance, and pulmonary hemodynamics in end-stage CKD patients on maintenance hemodialysis with pre-capillary PH was investigated in this study.

## 2. Methods

This retrospective observational study was approved by the Ethics Review Committee of Kyushu University Hospital (approval number: 29-526). The requirement of informed consent from patients was waived because of the retrospective observational study design. We reviewed medical records from our hospital between July 2014 and November 2020 and identified 393 patients with suspected PH who underwent right heart catheterization. Of these, seven patients who were on hemodialysis and had isolated pre-capillary PH or Cpc-PH were selected. The day after hemodialysis, a right heart catheterization was performed on these patients. Pre-capillary PH or Cpc-PH was defined as PH with a pulmonary vascular resistance of ≥3.0 Wood units and a mean pulmonary arterial pressure of >20 mmHg, according to the recommendations of the 6th World Symposium on Pulmonary Hypertension [17]. Age, sex, medical history, primary cause of kidney disease, and systolic and diastolic left ventricular function on echocardiogram were among the demographic and clinical data that were collected. All seven patients with isolated PH or Cpc-PH were treated with pulmonary vasodilators, including prostacyclin (an endothelin receptor antagonist) and phosphodiesterase 5 inhibitor. The treatment strategy is explained in the next section. Following this, we compared the World Health Organization (WHO, Geneva, Switzerland) functional class, six-minute walk distance, catheter-based pulmonary hemodynamics, heart rate, and mean systemic blood pressure at baseline and after treatment with pulmonary vasodilators.

### 2.1. Prevailing Treatment Strategy for Hemodialysis Patients with Pre-Capillary PH

The clinical management for pre-capillary PH in patients on hemodialysis was based on the guideline-directed medical therapy for idiopathic/heritable pulmonary arterial hypertension [18]. Pulmonary vasodilator therapy was initiated using a single drug that was carefully up-titrated by paying attention to hypotension and pulmonary congestion. If risk stratification of pulmonary arterial hypertension did not satisfy the low-risk profile, then another type of pulmonary vasodilator was added [18]. After 3–6 months of treatment, the WHO functional class, six-minute walk distance, and catheter-based pulmonary hemodynamics were re-evaluated.

### 2.2. Statistical Analysis

Data are presented as medians and interquartile ranges (IQRs). Continuous variables were tested using a paired t-test, whereas categorical variables were tested using the chi-square test. A *p*-value of <0.05 was considered statistically significant. Statistical analyses were performed using Statcel software, version 3.

## 3. Results

Of 393 patients with PH, 7 met the diagnosis of pre-capillary PH with end-stage CKD on hemodialysis. No patients with Cpc-PH were found [17,19]. The demographic and clinical data of individual patients are shown in Table 1 and Appendix A. Of the seven patients, four were males and three were females with a median age of 62 years (IQR, 59–76). Diabetic nephropathy was the most common etiology of CKD. The mean duration from the initiation of hemodialysis to the onset of pre-capillary PH was 6 years (IQR, 5–17). A history of percutaneous coronary intervention was found in two patients. One patient had atrial fibrillation. None of the patients had a history of myocardial infarction, valvular disease, intracardiac shunt, interstitial lung disease, or chronic obstructive pulmonary disease. The left ventricular ejection fraction was preserved in all patients (69% (IQR, 64–72%)). Based on the American Society of Echocardiography statement [20], left ventricular diastolic dysfunction was impaired in all patients (E/e’: 17 (IQR, 15–19), e’: 4.5 cm/sec (IQR, 4.1–5.0), left atrial volume index: 40 mL/m^2^ (IQR, 36–44), tricuspid regurgitant velocity: 3.8 m/sec (IQR, 3.6–3.8)).

The observation period from baseline to follow-up was 7 months (IQR, 6–27). None of the patients had anemia or infection before treatment. The low-risk profile improved with monotherapy in three patients (43%). However, the other three patients required dual or triple combination therapy; one patient’s profile improved to the low-risk profile. Figure 1 shows several treatment outcome measures with pulmonary vasodilators. Dry weight did not change significantly (baseline: 52 kg (IQR, 57–54), follow-up: 50 kg (IQR, 46–53); *p* = 0.36, *n* = 7). Plasma B-type natriuretic peptide levels decreased in almost all patients from 719 pg/mL (IQR, 631–1097) at baseline to 238 pg/mL (IQR, 92–521) during follow-up (*p* = 0.07, *n* = 6). The percentage of patients with WHO functional class I or II increased from 0% at baseline to 100% during follow-up (*p* = 0.0002, *n* = 7). Moreover, six-minute walk distance significantly increased from 273 m (IQR, 185–365) at baseline to 490 m (IQR, 470–550) during follow-up (*p* = 0.03, *n* = 4). 

Figure 2 shows the changes in hemodynamics after treatment with pulmonary vasodilator compared to baseline. All patients who were taking antihypertensive drugs discontinued them. There was no significant change in mean systemic blood pressure from 90 mmHg (IQR, 72–101) at baseline to 90 mmHg (IQR, 68–93) during follow-up (*p* = 0.50, *n* = 7). Resting heart rate decreased significantly from 88 bpm (IQR, 76–90) at baseline to 69 bpm (IQR, 67–77) during follow-up (*p* = 0.04, *n* = 7). Resting cardiac index increased from 2.9 L/min/m^2^ (IQR, 2.5–3.1) at baseline to 3.2 L/min/m^2^ (IQR, 3.2–3.6) during follow-up (*p* = 0.11, *n* = 7); however, it did not reach statistical significance. Pulmonary vascular resistance decreased significantly from 5.9 Wood units (IQR, 5.5–7.6) at baseline to 3.1 Wood units during follow-up (IQR, 2.6–3.3) (*p* = 0.02, *n* = 7). Pulmonary artery wedge pressure increased from 10 mmHg (IQR, 7–11) at baseline to 11 mmHg (IQR, 8–16) during follow-up (*p* = 0.04, *n* = 7). Therefore, pulmonary artery pressure did not change significantly; it changed from 35 mmHg (IQR, 33–37) at baseline to 32 mmHg (IQR, 25–36) during follow-up (*p* = 0.13, *n* = 7).

## 4. Discussion

To the best of our knowledge, this is the first case series demonstrating the effect of pulmonary vasodilators on changes in symptoms, exercise tolerance, and pulmonary hemodynamics in pre-capillary PH patients with end-stage CKD on maintenance hemodialysis. The results of this study showed that pulmonary vasodilators improved the symptoms, exercise tolerance, and pulmonary vascular resistance. Moreover, pulmonary vasodilators improved the six-minute walk distance and WHO functional class, although there were no changes in mean pulmonary artery pressure.

### 4.1. Effects of Pulmonary Vasodilators on Hemodynamics and Exercise Tolerance

Pulmonary vasodilators for PH in patients with CKD on hemodialysis decreased pulmonary vascular resistance and improved exercise tolerance. However, pulmonary vasodilators were ineffective in Case 7, whose underlying disease was type 1 glycogen storage disease. Glycogen storage disease causes CKD and pulmonary artery hypertension because of glycogen deposition in multiple organs. It also suggests that Case 7 was not primarily the case of hemodialysis-related pulmonary artery hypertension. Further, the effect of pulmonary vasodilators might depend on a primary underlying disease.

The increase in the cardiac index by pulmonary vasodilators might explain the improvement in exercise tolerance and symptoms in the patients included in our study. The decrease in the resting heart rate after treatment with pulmonary vasodilators suggested increased stroke volume and improved left ventricular filling following the right–left volume shift. In the present case series, antihypertensives were discontinued during treatment with pulmonary vasodilators in all patients, and systemic blood pressure did not change after treatment compared with baseline. The hypotensive effect is a common adverse effect of pulmonary vasodilators. As systemic hypotension causes dysdialysis, dose readjustment or discontinuation of antihypertensives should be considered after the initiation of pulmonary vasodilators. To avoid the adverse hypotensive effect of pulmonary vasodilators, it is reasonable to consider discontinuing antihypertensive drugs in routine clinical practice. Pulmonary vasodilators increased pulmonary capillary wedge pressure unexpectedly in this series, possibly due to a right-to-left volume shift. Even if the dry weight remains unchanged, appropriate blood volume control by hemodialysis and its reduction (in most cases) is needed when starting or increasing doses of pulmonary vasodilators.

### 4.2. Comorbid Left Ventricular Diastolic Dysfunction in Patients on Hemodialysis Is a Pitfall in Treating Existing Pre-Capillary PH

Multiple factors are involved in the development of PH in patients on hemodialysis. Therefore, PH associated with hemodialysis is classified as group 5 (unclear or multifactorial mechanisms) of the NICE classification [18]. Edmonston et al. reported that post-capillary PH and Cpc-PH are the most common subtypes in patients with end-stage CKD on hemodialysis [16]. The current study found no Cpc-PH, probably due to strict fluid control before right-heart catheterization. We focused on pre-capillary PH and evaluated the effects of pulmonary vasodilators on symptoms, exercise tolerance, and hemodynamics. This study suggests that sequential combination therapy with pulmonary vasodilators safely and effectively improves symptoms, exercise tolerance, and pulmonary vascular resistance in hemodialysis patients with pre-capillary PH. However, post-capillary PH or Cpc-PH developed in almost all patients at the post-treatment evaluation. Most patients on hemodialysis have subclinical left ventricular diastolic dysfunction, such as left ventricular hypertrophy. Underlying diastolic left ventricular dysfunction and insufficient fluid management after drug administration increased pulmonary capillary wedge pressure. Fortunately, there were no patients with overt pulmonary edema, although all patients in this study had diastolic dysfunction. However, this research suggests that pulmonary vasodilators can increase the risk of pulmonary edema. To avoid pulmonary edema, pulmonary vasodilators should be initiated in a dehydrated state following hemodialysis, with careful monitoring by chest X-ray or echocardiography. Following the use of pulmonary vasodilators, cardiac catheterization is mandatory. The increased prevalence of left ventricular diastolic dysfunction and the right-to-left volume shift in hemodialysis patients may be a barrier when using pulmonary vasodilators for pre-capillary PH.

### 4.3. Treatment Algorithm for Pre-Capillary PH in Patients on Hemodialysis

Based on the latest recommended treatment algorithm for pulmonary hypertension from the 6th World Symposium on Pulmonary Hypertension [21], we have modified the previously reported treatment strategy for PH in dialysis patients and propose our treatment strategy in Figure 3. Patients with anemia, infection, or other conditions that may cause a high cardiac output state are highly susceptible to pulmonary edema. Such conditions must be treated before starting pulmonary vasodilators. While selecting pulmonary vasodilators, drug metabolism should be considered. Riociguat (a soluble guanylate cyclase stimulator) and tadalafil (a phosphodiesterase-5 inhibitor) are contraindicated in patients on hemodialysis. Prostacyclin (an endothelin receptor antagonist) and sildenafil are other alternatives. To minimize pulmonary edema and dialysis-related hypotension, we recommend starting with oral monotherapy rather than dual combination therapy and gradually upregulating the dose. Although an indwelling venous catheter poses a risk of catheter infection, especially in patients on hemodialysis, subcutaneous or intravenous prostacyclin may be used in patients with a high-risk profile. Sequential add-on therapy should be considered until risk stratification reaches a low-risk profile with special attention to pulmonary edema and dysdialysis. 

## 5. Limitation

This was a retrospective observational study with a small sample size. Hence, large-scale, prospective randomized controlled trials are needed to confirm the efficacy of our proposed treatment strategy.

## 6. Conclusions

This case series shows that pulmonary vasodilators improve symptoms, exercise tolerance, and pulmonary hemodynamics in patients on hemodialysis with pre-capillary PH or Cpc-PH. Further, careful observation for post-capillary PH is required during up-titration of pulmonary vasodilators.

## Figures and Tables

**Figure 1 life-12-00780-f001:**
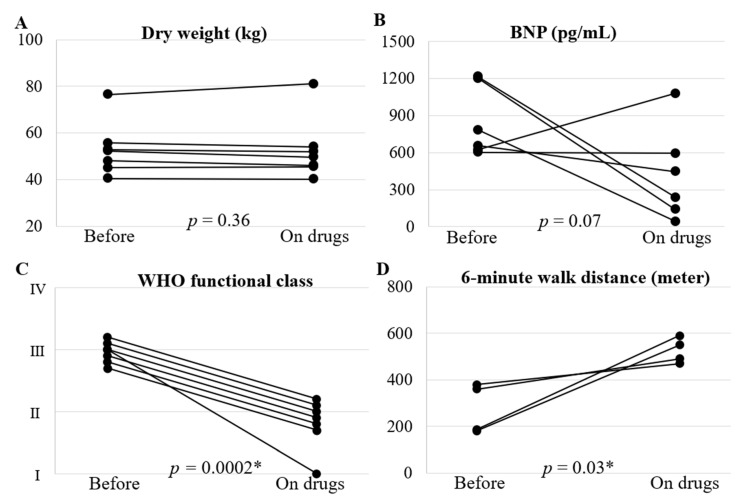
Treatment outcome measures with pulmonary vasodilators. The panels show changes in (**A**) dry weight, (**B**) BNP, (**C**) WHO functional class, and (**D**) 6-minute walk distance from before to after treatment with pulmonary vasodilators. The median duration from baseline to post-treatment evaluation was 7 months (IQR, 6–27). *: *p* < 0.05. WHO, World Health Organization; BNP, B type natriuretic peptide.

**Figure 2 life-12-00780-f002:**
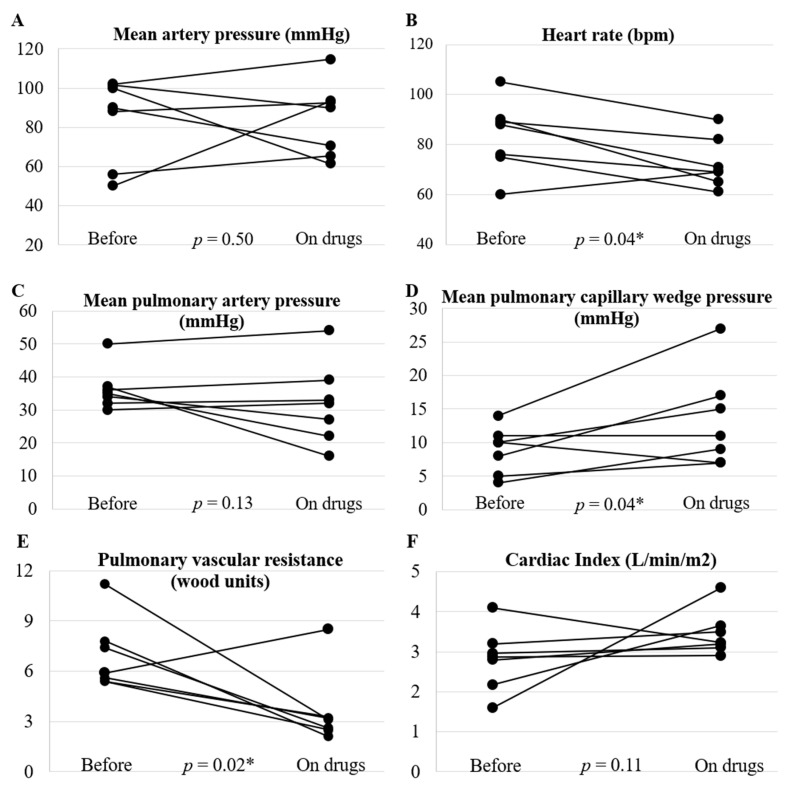
Hemodynamic changes before and after treatment with pulmonary vasodilators. The panels show changes in (**A**) mean systemic blood pressure, (**B**) heart rate and catheter-based pulmonary hemodynamics, (**C**) mean pulmonary artery pressure, (**D**) mean pulmonary capillary wedge pressure, (**E**) pulmonary vascular resistance, and (**F**) cardiac index. The median duration from baseline to post-treatment evaluation was 7 months (IQR, 6–27). *: *p* < 0.05.

**Figure 3 life-12-00780-f003:**
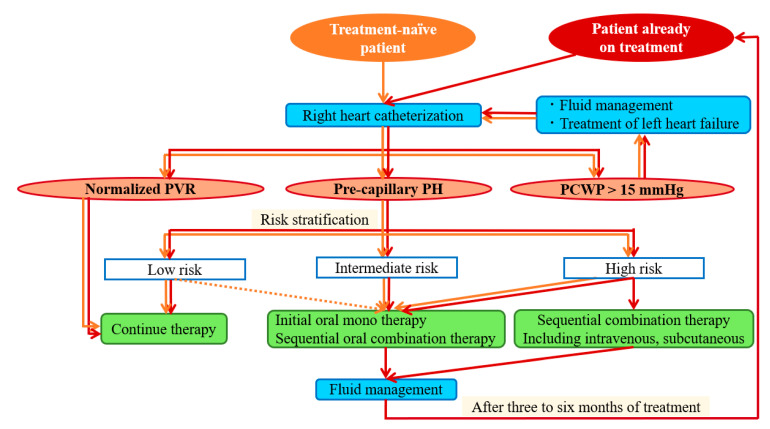
Treatment strategies for hemodialysis patients with pre-capillary PH. The panel shows a treatment algorithm in both treatment naïve and on treatment patients. PVR, pulmonary vascular resistance; PH, pulmonary hypertension; PCWP, pulmonary capillary wedge pressure.

**Table 1 life-12-00780-t001:** Demographic and clinical data of the patients included in the study (*n* = 7).

Patient Number	1	2	3	4	5	6	7
** *Patient Characteristics* **							
**Age, years**	52	73	61	78	62	83	56
**Sex**	Male	Female	Female	Male	Male	Male	Female
**Body mass index, kg/m^2^**	26.5	19.0	17.2	22.8	17.4	19.3	19.0
** *Kidney Disease and Hemodialysis* **							
**Primary kidney disease**	Diabetic nephropathy	Chronic glomerulonephritis	Lupus nephritis	IgA nephropathy	Horseshoe kidney	Diabetic nephropathy	Glycogen storage disease
**Duration of dialysis, years**	6	4	24	14	19	6	3
** *Past Medical History and Comorbidity* **							
**Prior pulmonary thromboembolism**	No	No	No	No	No	No	No
**Prior myocardial infarction**	No	No	No	No	No	No	No
**Coronary artery disease**	No	No	No	Yes	No	Yes	No
**Valvular heart disease**	No	No	No	No	No	No	No
**Intracardiac shunt**	No	No	No	No	No	No	No
**Atrial fibrillation/Atrial flutter**	No	No	No	No	No	Yes	No
**Interstitial lung disease**	No	No	No	No	No	No	No
**Chronic obstructive pulmonary disease**	No	No	No	No	No	No	No
**Hypertension**	Yes	Yes	Yes	No	Yes	No	Yes
**Diabetes mellitus**	Yes	No	No	No	No	Yes	No
** *Echocardiographic Parameters* **							
**LVEF, %**	69	75	74	62	70	66	61
**E/e’**	16.8	19.5	31.5	9.4	17.8	14.9	18.6
**e’, cm/sec**	4.3	5.1	4.8	3.9	4.5	5.2	3.6
**Left atrial volume index, mL/m^2^**	N/A	N/A	53.0	41.5	37.6	N/A	47.0
**Tricuspid regurgitant velocity, m/s**	3.8	3.2	3.8	3.5	3.8	3.7	4.3
** *Treatment* **							
**Pulmonary vasodilator**	Bosentan 62.5 mg b.i.d., Sildenafil 10 mg t.i.d.	Macitentan 10 mg q.d., Sildenafil 20 mg t.i.d.	Sildenafil 10 mg t.i.d.	Selexipag 1.0 mg b.i.d.	Treprostinil (subcutaneous) 50 ng/kg/min, Sildenafil 10 mg t.i.d., Bosentan 62.5 mg b.i.d.	Selexipag 2.0 mg b.i.d.	Ambrisentan 5.0 mg q.d., Sildenafil 10 mg b.i.d., Selexipag 0.8 mg b.i.d.
**Observation period from baseline to post-treatment evaluation, months**	8	7	6	6	45	6	108

LVEF, left ventricular ejection fraction; q.d., once daily; b.i.d., twice daily; t.i.d., three times daily; e’, mitral annular early diastolic velocity; E/e’, ratio between early mitral inflow velocity and mitral annular early diastolic velocity.

## Data Availability

Not applicable.

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
