# Peer review of "Beneficial Effects of Pulmonary Vasodilators on Pre-Capillary Pulmonary Hypertension in Patients with Chronic Kidney Disease on Hemodialysis"

_life, 2022, doi:10.3390/life12060780_

Round 1
Reviewer 1 Report
Kimuro et al. evaluated the effect of pulmonary vasodilators in patients with pulmonary hypertension and end-stage CKD on hemodialysis. This is interesting case series and well executed, but following issues need to be responded.
Major concerns
- Recently, Edmonston et al. showed the prevalence of different PH subtypes and their association with all-cause mortality in patients with CKD including hemodialysis patients (Am J Kidney Dis 2020). The data showed hemodynamic definition of PH was important predictor in patients with PH and CKD. Although the authors in this manuscript mentioned the importance of the definition in the introduction and discussion, the case series does not describe the hemodynamic typing of PH, so we do not know what proportion of patients had Cpc-PH or how many had pre-capillary PH, nor do we know the drug responsiveness of each pathology. Based on such data, it would be presumptuous to describe a treatment strategy as shown in Figure 3. Please show diastolic pressure gradient; DPG of each patient and discuss.
- Interstitial lung disease also significantly affects pulmonary vascular resistance. Please also describe the presence or absence of interstitial lung disease complications.
- In the methods, the definition of pulmonary hypertension states that the mean pulmonary arterial pressure is higher than 15 mmHg, but I think it may be a mistake for 20 mmHg.
Reviewer 2 Report
Kimura et al provided a case series of CKD on hemodialysis with pulmonary hypertension treated with pulmonary vasodilators. With their research authors provide an insight on a less explored are of PH. My only suggestion would be on presenting data. they provided 7 patients data which means it is impossible to be normally distributed. So it would be better to represent this data as median and IQR when appropriate. Also decimal points may not be necessary when representing this kind of data.
Best regards
Round 2
Reviewer 1 Report
The authors addressed my concerns.